# Kukoamine B from Lycii Radicis Cortex Protects Human Keratinocyte HaCaT Cells through Covalent Modification by *Trans*-2-Nonenal

**DOI:** 10.3390/plants12010163

**Published:** 2022-12-29

**Authors:** Hye Mi Kim, Jae Yong Kim, Ji Hoon Kim, Chul Young Kim

**Affiliations:** College of Pharmacy and Institute of Pharmaceutical Science and Technology, Hanyang University, Ansan 15588, Gyeonggi-do, Republic of Korea

**Keywords:** kukoamine B, Lycii Radicis Cortex, *trans*-2-nonenal, keratinocytes, pyridinium derivatives

## Abstract

The unsaturated aldehyde *trans*-2-nonenal is known to be generated by lipid peroxidation at the surface of the skin in an aging-related manner and has harmful effects on keratinocytes in the skin. In this study, the protective effect of a Lycii Radicis Cortex (LRC) extract against *trans*-2-nonenal-induced cell damage on human keratinocyte cell lines (HaCaT) was investigated. Notably, treatment with the LRC extract resulted in an increase in cell survival, while *trans*-2-nonenal decreased the viability of HaCaT cells. For identification of interaction between the LRC extract and *trans*-2-nonenal, this mixture was incubated in simulated physiological conditions, showing a strong decrease in the amount of *trans*-2-nonenal by the LRC extract. Subsequent LC-ESI-MS analysis revealed that kukoamine B (KB) formed Schiff base-derived pyridinium adducts with *trans*-2-nonenal. Thus, these results suggest that KB could be a potential agent that may protect HaCaT cells by forming new products with *trans*-2-nonenal.

## 1. Introduction

Lipid peroxidation represents a degradative process in the body arising from the production and propagation of free radical reactions primarily involving membrane polyunsaturated fatty acids [1]. This leads to the formation of a variety of lipid aldehydes including *α*,*β*-unsaturated aldehydes, di-aldehydes, and keto-aldehydes. These lipid aldehydes have been implicated in the pathogenesis of many diseases, including metabolic, cardiovascular, and inflammatory disease [2]. Specifically, because of the presence of several electrophilic reaction sites, these reactive aldehydes readily react with protein, DNA, and lipid to form covalent adducts, leading to the disruption of important biological functions. Of these, *α*,*β*-unsaturated aldehydes, such as 2-alkenals and 4-hydroxy-2-alkenals, are important agents for covalent modification of reacted molecules.

Among the 2-alkenals, *trans*-2-nonenal has a characteristically unpleasant greasy and grassy odor endogenously generated during lipid peroxidation as a minor product [3]. This is known to be generated by aging and characterized as the key odor component in elderly people [4]. *trans*-2-nonenal is considered an electrophilic unsaturated aldehyde containing two electrophilic reaction centers and, therefore, exhibits high reactivity toward nucleophiles such as thiol or amino compounds [5]. It was previously shown that *trans*-2-nonenal covalently modified through reaction of aldehyde with lysine side-chain amino groups and 2-nonenal-lysine adducts was confirmed during lipid peroxidation-mediated modification of protein in vitro and in vivo [6]. Furthermore, *trans*-2-nonenal was injurious to keratinocytes through promoted apoptosis and reduced the thickness and the number of proliferative cells in a three-dimensional epidermal equivalent model [7]. Therefore, we hypothesize that the protective effects against *trans*-2-nonenal on keratinocytes might involve a process of chemical inactivation through its structural modification. To test this idea, plant extracts were screened to evaluate whether the effect of *trans*-2-nonenal on cultured human epidermal keratinocytes (HaCaT cells) was modified.

Lycii Radicis Cortex (LRC), the root bark of *Lycium chinense* Mill., is termed *jigolpi* in Korean. LRC is used as a cooling agent to “clear heat” in East Asian traditional medicine and exerts functions in some chronic diseases such as cough, hypertension, and diabetes [8]. Recently, LRC has been reported to have many pharmacological effects such as anti-osteoporotic [9], anti-inflammatory [10,11], anti-oxidant [12], anti-depressant [13], anti-tumor [14,15], and blood glucose regulation [16] effects. Previous phytochemical investigations of LRC identified the presence of alkaloids, flavonoids, phenolic acids, coumarins, and other compounds [8]. Among these constituents, kukoamines A and B (KA and KB, respectively) were identified as unique markers for LRC because they are truly bioactivity-related, genus-specific, and content-abundant in this herb [17].

In the present study, to understand the mechanism underlying the formation of a covalently modified active compound in LRC extract with *trans*-2-nonenal, we investigated the efficacy of quenching with *trans*-2-nonenal and identified new adducts specifically generated in LRC extract modified by *trans*-2-nonenal.

## 2. Results and Discussion

### 2.1. Protective Effects of LRC Extract on Human Keratinocytes through Quenching Activity with Trans-2-Nonenal

Because *trans*-2-nonenal are injurious to keratinocytes, as previously described, we examined whether a protective effect of LRC extract against the effect of *trans*-2-nonenal occurs in keratinocytes. After the LRC extract was treated to the HaCaT cells, the viability of the cells was then measured with the WST-8 assay. The results show that LRC extract treatment does not affect the viability of the keratinocytes (Appendix A). As shown in Figure 1A, *trans*-2-nonenal significantly reduced the viability of keratinocytes, exhibiting 55 % (*p* < 0.001). After the cells were treated with LRC extract and *trans*-2-nonenal, the results showed that LRC extract treatment provided significant protection at 10 μg/mL These results suggest that the effects against *trans*-2-nonenal, which decreased the viability of HaCaT cells, were restored by the LRC extract.

To identify the interaction between the LRC extract and *trans*-2-nonenal, the LRC extract (1, 10, 100 mg/mL) was exposed to *trans*-2-nonenal (0.1 mM) under physiological conditions. Changes in the amount of *trans*-2-nonenal were then examined by HPLC analysis. As shown in Figure 1B, it was revealed that the remaining amount of *trans*-2-nonenal significantly decreased with the increase in LRC extract concentration. Notably, 100 mg/mL of the LRC extract demonstrated the highest activity, with over 90 % of *trans*-2-nonenal removed. These results clearly reveal the protective effects of the LRC extract on keratinocytes, demonstrated by removing *trans*-2-nonenal through the quenching activity of LRC extract.

### 2.2. Screening of the Trans-2-Nonenal Quenching Components in LRC Extract

In view of the chemical structure, *trans*-2-nonenal displays different reactivity by either reacting with the electrophilic double bond or giving rise to a nucleophilic addition on the carbonyl group to form a Schiff base [5]. These two electrophilic reaction centers can react with the target compound to form a new adduct, so such a compound is considered to possess quenching activity against *trans*-2-nonenal. On this basis, the amounts of both *trans*-2-nonenal and the target compound decrease once these compounds react, while the new adduct increases accordingly.

In the present study, the LRC extract and the reaction mixture after incubation with *trans*-2-nonenal was analyzed by HPLC for comparison. To optimize the experiment parameters, the LRC extract was incubated with various concentrations of *trans*-2-nonenal for different incubation times. As shown in Appendix A, the amount of both *trans*-2-nonenal and the target compound (e.g., KB and feruloylputrescine) significantly decreased or even disappeared with the increase in concentration of *trans*-2-nonenal and incubation time, while four product peaks increased accordingly. The LRC extract was incubated with 200 mM of *trans*-2-nonenal for 72 h to ensure the discovery of a potential candidate with quenching activity in this study (Figure 2). KB and feruloylputrescine in the LRC extract were regarded as the potential quencher of *trans*-2-nonenal.

Among these potential candidates, the HPLC chromatogram showed the major peak of constituents from the LRC extract, and it was identified as KB, by comparing its retention time, UV, and MS with the reference standard compound (Appendix A). Next, the quantitative analysis of KB in the LRC extract was performed to identify the major component as KB using HPLC. As shown in Appendix A, the content of KB in the LRC extract, calculated by a standard curve, was 4.26 %.

### 2.3. Effects of KB on Human Keratinocytes trhough Formation of the Trans-2-Nonenal Adducts

KB was identified as one of the representative compounds in the LRC extract, and found to possess especially outstanding anti-oxidation and anti-inflammation properties, which were closely related to the “heat cleansing” function of the LRC [18,19,20]. Because it was revealed that KB might be one of the active compounds against *trans*-2-nonenal in this study, KB was evaluated for block effects, where *trans*-2-nonenal decreases the cell viability of HaCaT cells. As shown in Figure 3A, the treatment of KB was effective in a dose-dependent manner.

For further identification of four product peaks generated from this reaction in the HPLC chromatogram (Figure 2), the adducts were analyzed with LC-ESI/MS. Among these peaks, two new peaks (t_R_ 38.01 and 38.29 min) had molecular ions at *m*/*z* 773.5, suggesting that these products were di-nonenal conjugated KB (Figure 3B, Appendix A). An additional two peaks appeared at 40.60 and 40.99 min, which had molecular ions at *m*/*z* 507.4, indicating that these peaks were di-nonenal adducts of feruloylputrescine (Figure 3C, Appendix A), which has been reported previously in LRC [21].

Although KA and KB, which are actually isomers of each other, were previously mentioned as unique markers in the LRC extract, their quenching activity against *trans*-2-nonenal was completely different. They are spermine alkaloids, and the only difference between these two compounds is the position of dihydrocaffeoylation. Among these compounds, KA has a linear-chain structure, in which two dihydrocaffeoyl moieties are individually connected to two terminal N-atoms; as for KB, one of the two dihydrocaffeoyl moieties is linked to one middle N-atom, which forms a branched chain structure [19]. Based on chemical characterization, the observed adducts at *m*/*z* 773.5 resulted from the Schiff base formation of *trans*-2-nonenal with the ε-amino group of KB. The Schiff base adduct may further be attacked by the enolate anion of the second aldehyde moiety, followed by dehydration and cyclization to form the final pyridinium derivatives (Figure 4). As such, the observed adducts at *m*/*z* 507.5 were also identified as the Schiff base-derived pyridinium adducts because the chemical characterization of feruloylputrescine is similar to KB. In addition, the Schiff base-derived adducts showed two peaks in the chromatogram (Figure 2), which are *cis*- and *trans*-derivatives. Based on previous studies [6], the formation of two isomeric pyridinium derivatives with *trans*-2-nonenal as the reactant may be due to isomerization of the *trans*-2-nonenal in the presence of amines.

## 3. Materials and Methods

### 3.1. Chemicals and Reagents

We purchased *trans*-2-nonenal (97%) from TCI (Japan). Kukoamine B was purchased from ChemFaces (Wuhan, China). High-performance liquid chromatography (HPLC) grade acetonitrile and water were purchased from Daejung Chemical (Gyonggi-do, Republic of Korea), and analytical grade formic acid and trifluoroacetic acid were purchased from Sigma-Aldrich (St. Louis, MO, USA).

### 3.2. Preparation of LRC Extract

Lycii Radicis Cortex (LRC) was purchased from an oriental herbal market in Kyungdong, Seoul, Republic of Korea. The root bark (684 g) was ground into powder and extracted with 50% aqueous ethanol for 1–4 h using an ultrasonic apparatus. The solution was concentrated by a rotary evaporator and freeze dried to yield 90.34 g of crude extract and then stored at −20 °C until required.

### 3.3. Cell Culture

HaCaT cells, which are human keratinocyte lines, were purchased from the American Type Culture Collection (ATCC, VA, USA). The cells were cultured in high glucose Dulbecco’s Modified Eagle’s Medium (DMEM) supplemented with 10% fetal bovine serum (FBS) and 1% penicillin-streptomycin (Gibco, UK) at 37 °C in a humidified 5% CO_2_ atmosphere. The cells were subcultured every 2 days, passaged three times, and then used for experiments.

### 3.4. Cell Viability Assay

The cell viability was examined using the WST-8 assay (Dojindo Laboratories, Kumamoto, Japan). HaCaT cells were seeded at a density of 4 × 10^4^ cells/well in a 48-well plate and incubated for 24 h. After the samples were treated (or with *trans*-2-nonenal) with a serum free medium for 24 h, they grew to be approximately 90% confluent. Thereafter, 20 μL of the WST-8 solution was added to the wells for 3 h. The absorbance was measured at 450 nm using a microplate reader.

### 3.5. Incubation of Trans-2-Nonenal with LRC Extract

Dried LRC extract was pre-dissolved in phosphate-buffered saline (PBS, 100 mM, pH 7.4) solution and extracted in an ultrasonic water bath for 2 h to obtain a sample solution. The different concentrations (1–100 mg/mL) of the extracted solution were incubated with 0.1 mM of *trans*-2-nonenal in a stirring water-bath at 37.5 °C for 24 h. After incubation, the reaction mixtures were analyzed by a reverse-phase HPLC to confirm the existence of residual *trans*-2-nonenal.

### 3.6. Quantification of Residual Trans-2-Nonenal by HPLC Analysis

Analytical HPLC was conducted using the Agilent 1260 HPLC system with Capcell pak C18 UG120 (4.6 mm i.d. × 250 mm, 5 µm, Shiseido, Japan). The mobile phase consisted of acetonitrile containing 0.1 % formic acid (solvent A) and water containing 0.1% formic acid (solvent B) with gradient elution (linear gradient from 10% to 100% A in 30 min). The flow rate was 1 mL/min, and a quantitative analysis was performed at 226 nm. All analyses were performed in triplicate.

### 3.7. Identification of Reaction Products by HPLC-DAD-ESI/MS Analysis

Liquid chromatography coupled with electrospray ionization-based mass spectrometry (LC-ESI-MS) analysis was performed on an instrument equipped with an electrospray ionization source interfaced to an Advion expression CMS mass spectrometer (Advion, Ithaca, NY, USA). Liquid chromatography was performed on a Waters 2695 Alliance system and carried out on a Cosmosil C18 column (2.0 mm i.d. × 150 mm, 5 µm). The mobile phase was composed of 0.1% trifluoroacetic acid water (solvent A) and acetonitrile (solvent B) with a gradient elution (0–5 min, 5–10% B, 5–25 min, 10% B, 25–45 min, 10–100% B, 45–50 min, 100% B). The mass operational parameters in positive ion mode were as follows: scan range, *m*/*z* 100–1200; capillary temperature, 250 °C; capillary voltage, 150 V; source voltage offset, 30; source voltage span, 10; source gas temperature, 120 °C; and ESI voltage, 3.5 kV.

### 3.8. Quantitative Analysis of KB in LRC Extract

Based on the HPLC-DAD analysis, the quantitative method of KB in the LRC extract was validated in terms of linearity, quantification (LODs and LOQs), and recovery test.

Linearity was examined with standard solutions. Five concentrations for KB were prepared with 0.1–2.5 mg/mL in methanol for the standard curve. Each concentration of the standard was injected five times. The linear regression equations were calculated with y = ax ± b, where x was concentration and y was peak area of each flavonoid. Linearity was established by the coefficient of equation (R^2^). All calibration curves showed good linear regression (R^2^ > 0.9995) within test ranges. The LODs and LOQs under the present HPLC-DAD method were determined at signal-to-noise ratios (S/N) of 3 and 10, respectively. The LRC extract was weighed (10 mg), dissolved, and vortexed in 1 mL of methanol. For standard sample, the prepared extract was replicated by three times. The recovery % was calculated using the following equation: Recovery (%) = (amount found − original amount)/amount spiked.

## 4. Conclusions

The findings of this study revealed that the Lycii Radicis Cortex (LRC) extract partially restored the harmful effects of *trans*-2-nonenal on HaCaT cells. Because *trans*-2-nonenal is considered as a reactive aldehyde, we can assume that the active compound in the LRC extract may be the molecules that intervene in quenching the carbonyl species. Once reacted under physiological conditions, kukoamine B (KB), the main constituent of LRC extract, potently and efficiently removed toxic *trans*-2-nonenal, effectively possessing *trans*-2-nonenal-trapping capacity. KB was also found to have significant beneficial effects in keratinocytes prevention. Upon an investigation of the adducts produced through quenching with *trans*-2-nonenal, we identified the Schiff base-derived pyridinium adducts of KB and feruloylputrescine in the LRC extract. The results presented herein show that an LRC extract or KB has the potential to not only reduce characteristic aging odor through the process of chemical inactivation of *trans*-2-nonenal generated from lipid peroxidation but also may be effective as a candidate for skin protection.

## Figures and Tables

**Figure 1 plants-12-00163-f001:**
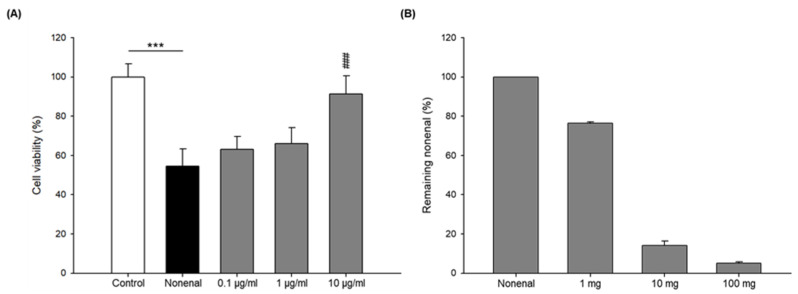
Effects of LRC extract with *trans*-2-nonenal on the cell viability of human keratinocytes and the quantitative reduction of *trans*-2-nonenal by LRC extract. These data are expressed as mean ± SD of three independent experiments. (**A**) Cell viability of human keratinocytes treated with *trans*-2-nonenal (150 μM) and LRC extract (0.1, 1, 10 μg/mL). *** *p* < 0.0001 vs. control; ### *p* < 0.0001 vs. *trans*-2-nonenal. (**B**) The LRC extract at different concentrations (1–100 mg/mL) and trans-2-nonenal (0.1 mM) were incubated in PBS (100 mM, pH 7.4) at 37.5 °C for 24 h.

**Figure 2 plants-12-00163-f002:**
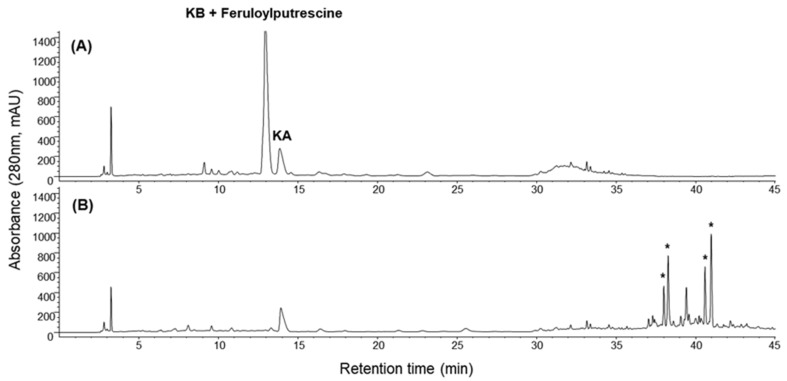
The HPLC chromatogram obtained for (**A**) the LRC extract (100 mg/mL) after incubation with (**B**) *trans*-2-nonenal in PBS (100 mM, pH 7.4) at 37.5 °C for 72 h. The peaks designated as asterisk (*) were new product peaks that appeared throughout this reaction.

**Figure 3 plants-12-00163-f003:**
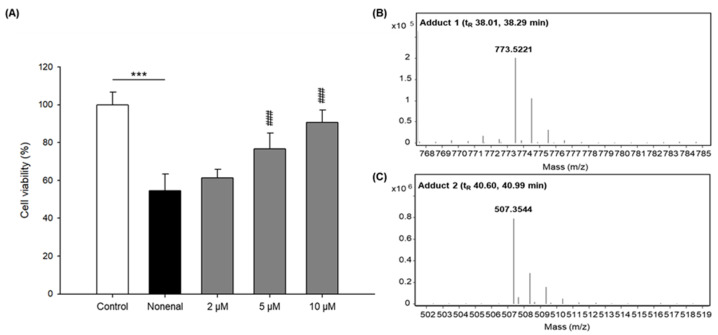
(**A**) Effects of KB with *trans*-2-nonenal on the cell viability of human keratinocytes and MS spectra of *trans*-2-nonenal adducts of (**B**) KB and (**C**) feruloylputrescine. Cell viability of human keratinocytes treated with *trans*-2-nonenal (150 μM) and KB (2, 5, 10 μM). These data are expressed as mean ± SD of three independent experiments. *** *p* < 0.0001 vs. control; ### *p* < 0.0001 vs. *trans*-2-nonenal.

**Figure 4 plants-12-00163-f004:**
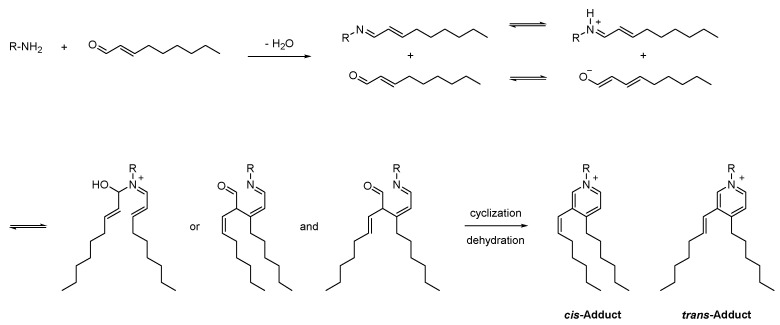
Proposed reaction mechanism leading to formation of pyridinium derivatives from amine-containing compounds in the presence of *trans*-2-nonenal.

## Data Availability

The data that support the findings of this study are available in the Appendix A of this article.

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
