# Peer review of "Kukoamine B from Lycii Radicis Cortex Protects Human Keratinocyte HaCaT Cells through Covalent Modification by Trans-2-Nonenal"

_plants, 2022, doi:10.3390/plants12010163_

Round 1

Reviewer 1 Report

In this paper effect of Lycii Radicis Cortex (LRC) extract against trans-2-nonenal-induced cell damage on human keratinocyte cell lines (HaCaT) was performed. The  interaction between the LRC extract and trans2-nonenal simulated physiological conditions were used, showing a strong decrease in the amount of trans-2-nonenal by LRC extract. LC-ESI-MS analysis revealed that kukoamine B (KB) formed Schiff base-derived pyridinium adducts with trans-2-nonenal. KB was showed to have significant beneficial effects in keratinocytes prevention

Comments please check accurately English language

2 please validate hplc ms quantitation by showing parameters such a lod loq, recovery, etc of main compounds

Reviewer 2 Report

This manuscript describes „Kukoamine B from Lycii Radicis Cortex protects human keratinocytes HaCaT cells through covalent modification by trans-2-nonenal” by Kim et al. and has been submitted as an article.

The main goal of the study was the identification of the interaction between the LRC extract and trans-2-nonenal, incubating the mixture in simulated physiological conditions, and observing a strong decrease in the amount of trans-2-nonenal by LRC extract. Subsequent LC-ESI-MS analysis revealed that kukoamine B (KB) formed Schiff base-derived pyridinium adducts with trans-2-nonenal. The above mentioned results suggest that KB in the plant extract could be a potential agent that may protect HaCaT cells. The authors also made proposed theoretical mechanism for this action. Since the therapeutic interest of the result, I can suggest the MS for acceptance with minor revision according to the next comments:

-          typing mistakes should be carefully checked and corrected like nonenal has many incorrect variations in the MS (see ex.: row 43, row 62 and row 84)

-          Figure 4 also has a typing mistake (cis-Addcuct)

-          Please give a quantitative information about the KB content of the extract was applied during the experiments!

-          the final page of Lit. 18. is missing (1843-1853)

-          Is there any information about the cation of the adducts determined by HPLC and MS?

The MS spectra of the four adducts (drawn also in the supporting info) would be great to see also in the supporting information.
